# Effect of Finite-Sized Optical Components and Pixels on Light-Field Imaging through Correlated Light

**DOI:** 10.3390/s22072778

**Published:** 2022-04-05

**Authors:** Gianlorenzo Massaro, Francesco Di Lena, Milena D’Angelo, Francesco V. Pepe

**Affiliations:** 1Dipartimento Interateneo di Fisica, Università degli Studi di Bari, 70126 Bari, Italy; milena.dangelo@uniba.it (M.D.); francesco.pepe@ba.infn.it (F.V.P.); 2Istituto Nazionale di Fisica Nucleare, Sezione di Bari, 70125 Bari, Italy; francesco.dilena@ba.infn.it

**Keywords:** light-field imaging, quantum imaging, correlation imaging, 3D imaging

## Abstract

Diffraction-limited light-field imaging has been recently achieved by exploiting light spatial correlations measured on two high-resolution detectors. As in conventional light-field imaging, the typical operations of refocusing and 3D reconstruction are based on ray tracing in a geometrical optics context, and are thus well defined in the ideal case, both conceptually and theoretically. However, some properties of the measured correlation function are influenced by experimental features such as the finite size of apertures, detectors, and pixels. In this work, we take into account realistic experimental conditions and analyze the resulting correlation function through theory and simulation. We also provide an expression to evaluate the pixel-limited resolution of the refocused images, as well as a strategy for eliminating artifacts introduced by the finite size of the optical elements.

## 1. Introduction

Light-field imaging is a rapidly developing technique that enables the simultaneous measurement of both light intensity distribution and propagation direction of light rays coming from a three-dimensional scene of interest [1]. The availability of the angular information represents the main difference between light-field (also known as *plenoptic*) and conventional imaging. The richness of the data collected by a light-field device entails the possibility to perform single-shot 3D sampling; to achieve the same result with a normal camera, multiple acquisitions on a stack of different planes would be needed [2,3,4]. Its scanning-free nature makes light-field imaging one of the fastest methods for 3D reconstruction and its applications span a large variety of fields, from photography [5,6,7,8] and microscopy [9] to real-time imaging of neuronal activity [10], wavefront sensing [11], and particle image velocimetry [12], to name a few. In its conventional form, light-field imaging is implemented by introducing an array of micro-lenses between the sensor and the imaging device (e.g., the camera lens); the role of the micro-lenses is to obtain a set of “sub-images” corresponding to different propagation directions. However, the micro-lenses impose a strong limitation on the image resolution, which cannot reach the diffraction limit and rapidly worsens with improved 3D reconstruction capabilities [3,13,14].

A recently developed approach to plenoptic imaging is based on retrieving the spatial and angular information by measuring light intensity correlations between two separate sensors. Therefore, the spatial and directional information can be retrieved with the maximum optical resolution, as set by wave-optics [15,16]. The technique, named *correlation plenoptic imaging* (CPI), does not need the micro-lenses array to decouple the propagation direction from the position of the light rays on the plane of interest, since this information is intrinsically encoded within the correlation function. CPI can thus produce images with the same diffraction-limited resolution as conventional imaging devices [16]. Furthermore, the measurement of plenoptic information through correlations has proven to better perform than the conventional approach, endowing CPI with unprecedented refocusing and depth-of-field extension capabilities. CPI is a quite versatile technique in that the idea of correlating two high spatial resolution detectors to obtain plenoptic information can be applied to any imaging technique, from photography to microscopy [17,18], and works, with the appropriate adaptations, with both thermal sources of light and entangled photons [19,20].

Regardless of the specific experimental realization, from the theoretical viewpoint, the workflow for decoding the plenoptic information from the CPI is unchanged: the correlation function is evaluated by modeling the propagation of the electromagnetic fields from the source to the sensors, a geometrical optics approximation is then performed to find the relationship between the position of rays impinging on the detectors and the coordinates on the imaged scene. The relevant parameters for the data processing algorithms are deduced from the expression of the correlation function in the ray optics regime. Interestingly, the variation of resolution with the axial position of the object is much slower in CPI than in conventional imaging [21] and has the peculiar property of being, to a large extent, independent of the numerical aperture of the imaging system [18]. In fact, only the resolution and the natural depth of field of the focused plane are determined by the numerical aperture of the imaging system; when planes outside of the natural depth-of-field are considered, the resolution of the resulting refocused image only depends on the illumination wavelength and the object size. In fact, the resolution in CPI is determined by the apertures when the object is focused, but depends only on the illumination wavelength and the object size as soon as one considers axial planes, which are outside of the natural depth-of-field of the lenses. In this article, for the first time, we outline the effect of both the finite extension of optical components and detectors, showing that large numerical apertures and detectors, although not influencing the resolution of the refocused images, are indeed beneficial for increasing the signal-to-noise ratio. Furthermore, the discrete nature of the measuring process (pixel size) will be considered, and the effect it has on the measured correlation function and on the refocusing capability of CPI will be investigated for the first time. Particularly, we shall show that the finite pixel size has non-trivial implications on the resolution (the specific expression will be provided) and affects the noise properties of the final image.

## 2. Materials and Methods

CPI is based on the simultaneous measurement and subsequent correlation of the light intensity distributions $I_{A}\left(\rho_{a}\right)$ and $I_{B}\left(\rho_{b}\right)$, where $\rho_{a}=\left(x_{a},y_{a}\right)$ and $\rho_{b}=\left(x_{b},y_{b}\right)$ are two-dimensional (transverse) space coordinates defined on the photosensitive planes of two detectors $D_{A}$ and $D_{B}$, respectively. The correlation function is evaluated from the measured intensity distributions as:(1)$$\Gamma^{\left(2\right)}\left(\rho_{a},\rho_{b}\right)=\langle I_{A}\left(\rho_{a}\right)I_{B}\left(\rho_{b}\right)\rangle -\kappa \langle I_{A}\left(\rho_{a}\right)\rangle \langle I_{B}\left(\rho_{b}\right)\rangle ,$$
where the angular brackets denote the statistical ensemble average [22] and the coefficient κ equals either 1, for thermal illumination, or 0, if the source is an emitter of entangled photon pairs [21,23]. Experimentally, an ergodic hypothesis is made on the statistical properties of the source, so as to replace the quantum state average with an operationally well-defined time average.

The dataset collected by a CPI apparatus is composed of a series of *N* pairs of frames, one for each detector, containing samples of the intensity impinging on the detectors surfaces at times $t_{1},t_{2},...,t_{N}$. Each *j*-th element of the series contains the 2D arrays:(2)$$I_{I}\left(\rho_{i}\right)=\int_{t_{j}}^{t_{j}+\Delta t}I_{I}\left(\rho_{i},t\right)dt,$$
where I=A,B; i=a,b; ***t*** denotes the time dependence of the intensity arriving on the detectors, and Δt is the exposure time, supposed to be constant throughout the measurement. Due to the discrete nature of pixels, the space variable $\rho_{i}$ is not continuous and must be considered as a discrete set of coordinates associated with the pixel centers. When a sufficient number of frames are collected, the discrete time average yields approximately the same results as the state average of Equation (Equation 1), so that, for κ=0,
(3)$$\Gamma_{\exp}^{\left(2\right)}\left(\rho_{a},\rho_{b}\right)=\frac{1}{N}\sum_{t\in \{t_{1},..,t_{N}\}}I_{A}\left(\rho_{a},t\right)I_{B}\left(\rho_{b},t\right)≃\Gamma^{\left(2\right)}\left(\rho_{a},\rho_{b}\right).$$

An analogous approximation holds for the case κ=1 in Equation (Equation 1). Each of the *N* terms $I_{A}\left(\rho_{a},t\right)I_{B}\left(\rho_{b},t\right)$ is a four-dimensional array obtained by computing the tensor product between the matrices $I_{A}$ and $I_{B}$ corresponding to the same acquisition time. The experimentally measured correlation function is thus a 4D array as well.

From now on, we shall assume that the sample of interest is a two-dimensional object characterized by a polarization-independent amplitude transmissivity $A(\rho_{s})$. In this case, the theoretical correlation function reads:(4)$$\Gamma^{\left(2\right)}\left(\rho_{a},\rho_{b}\right)={\left|\int_{-\infty }^{+\infty }A\left(\rho_{s}\right)\int_{-\infty }^{+\infty }A^{*}{\left(\rho_{s}^{\prime }\right)}^{m}\Phi \left(\rho_{s},\rho_{s}^{\prime },\rho_{a},\rho_{b}\right)d^{2}\rho_{s}^{\prime }\;d^{2}\rho_{s}\right|}^{2},$$
where $A^{*}$ denotes the complex conjugate of the amplitude transmissivity, and $\rho_{s}$ and $\rho_{s}^{\prime }$ are the transverse coordinates on the object plane [17,21,23]. The integer coefficient *m* and Φ depend on the particular realization of CPI. If the object is placed only in one of the two arms, as in Figure 1a, then m=0. Instead, if light crosses the object before it is split towards the detectors, as in Figure 1b, then m=1. The complex function Φ represents the point spread function of CPI and depends both on the light source, and on the configuration of the optical paths [17,21,23]. Within Φ are enclosed all the details about the field propagation through the optical components, and also the coherence properties of the source [24]. The expression of Φ is analyzed extensively in Refs. [17,21,23]. For our purpose, it is sufficient to consider it as a second-order transfer function [25], describing the (back-)propagation of a correlated detection at coordinates $\rho_{a}$ on $D_{A}$ and $\rho_{b}$ on $D_{B}$ towards the object plane. Let us now make a few assumptions and approximations that enable simplifying Equation (Equation 4) and relating the aperture function *A* to the coordinates on the detectors, namely, establishing a relationship between the two detectors and the object plane. We consider Φ in the paraxial approximation [25] and assume perfect mode-to-mode correlation [24] (i.e., infinitesimal transverse coherence area) on the source. Furthermore, all the apertures in the setup are assumed to be infinite (including the source transverse size), so that Φ has an analytical expression, and the two double integrals on the object plane are easily solved in the asymptotic limit of geometrical optics [21]; in this limit, the result of all integrals can be approximated by applying a stationary phase method [26]. Since all optical propagators in the paraxial approximation involve phases that are, at most, quadratic in the integration variables, all functions come out to depend on linear combinations of the detector variables. In fact, the correlation function reduces to:(5)$$\Gamma_{\mathrm{geom}}^{\left(2\right)}\left(\rho_{a},\rho_{b}\right)={\left|A(\alpha \left(z\right)\rho_{a}+\beta \left(z\right)\rho_{b})\right|}^{2(1+m)}.$$

Hence, regardless of the specific CPI configuration, the correlation function in the geometrical approximation reduces to the power 1+m of the object intensity transmissivity (the square module of the amplitude transmissivity that appears in Equation (Equation 4)), evaluated in a linear combination of $\rho_{a}$ and $\rho_{b}$. The coefficients α and β strictly depend on the experimental realization of CPI, and are a function of the axial position *z* of the object plane. Equation (Equation 5) thus shows a point-to-point correspondence between coordinates at the two detectors and areas of the object planes, which is determined only by the two coefficients α and β. This interpretation shows the convenience of the ray optics limit for interpreting the measured correlation function. Two more aspects of Equation (Equation 5) are worth emphasizing: first, the coordinates *x* and *y* are not mixed up by defocusing; second, α and β do not depend on the transverse direction, namely, they are the same for the two independent directions *x* and *y* on the detector planes.

To better understand the extent of the considerations above, it is worth noticing that, in the most general case, the 2D aperture function *A* could depend on a linear combination of the four coordinates at the detectors; thus, involving up to eight coefficients: four in the first argument involving all four components on the detectors, and four in the second argument. However, from Equation (Equation 5), we see that only two coefficients, α and β, are needed and the problem of refocusing can be solved along *x* and *y* separately.

For the sake of simplicity and ease of visualization, from now on, the second-order transfer function Φ shall be assumed factorable with respect the two transverse coordinates *x* and *y* (i.e., $\Phi \left(\rho_{s},\rho_{s}^{\prime },\rho_{a},\rho_{v}\right)=\Phi_{x}\left(x_{s},x_{s}^{\prime },x_{a},x_{b}\right)\Phi_{y}\left(y_{s},y_{s}^{\prime },y_{a},y_{b}\right)$) and symmetric in the two dimensions ($\Phi_{x}=\Phi_{y}$) throughout the rest of the paper, even outside of the geometrical approximation. These assumptions imply that field propagation from the source to the detectors has cylindrical symmetry around the optical axis, and also that the expression of propagators is such that integration along orthogonal transverse directions can be solved independently. In this context, the 2D treatment leads to the same results as the complete 4D case, with the added benefit of having 2D correlation functions that are easily represented as images. Moreover, the factorization and symmetry are generally good approximations of the experimental conditions. Based on these assumptions, we shall drop the boldface from space variables, regard the detectors as mono-dimensional photosensitive strips and the object transmissivity as a 1D function.

Going back to Equation (Equation 5), we see that the correlation function is an infinitely extended collection of shifted and magnified replicas of the object aperture function. In fact, for each given coordinate $x_{a}$ on $D_{A}$, the correlation function of Equation (Equation 5) represents a shifted and magnified “copy” of the object on $D_{B}$; the shift amounts to $-\alpha x_{a}$ and the magnification to 1/β. The analogous is obtained by fixing $x_{b}$. Another way to look at this is the following: a detail at coordinate $x_{s}$ in the object plane is infinitely replicated along the line $\gamma_{z}\left(x_{s}\right)$ of equation $\alpha \left(z\right)x_{a}+\beta \left(z\right)x_{b}=x_{s}$ in the space defined by the coordinates at the detectors, $\left(x_{a},x_{b}\right)$. This picture enables a very intuitive understanding of the process of refocusing, as well as of the role played by the coefficients α and β. In fact, the line integral involved in the refocusing process: (6)$$\Sigma_{z}\left(x_{s}\right)=\int_{\gamma_{z}\left(x_{s}\right)}\Gamma_{\exp}^{\left(2\right)}\left(x_{a},x_{b}\right)d\ell ;$$
enables summing together all the contributions in the correlation function to a single point of the object. The result is proportional to the intensity transmissivity profile of the object, evaluated in the single object coordinate $x_{s}$. By repeating this operation over the entire object plane, i.e., for all the possible $x_{s}$, the whole transmission function of the object is reconstructed, namely, the image is refocused. The orientation of the lines along which the integration of Equation (Equation 6) is performed is uniquely identified by the ratio α(z)/β(z), which, in turn, depends on the position *z* of the object along the optical axis. The dependence of α and β on the axial position indicates that, when refocusing on a different object plane $z^{\prime }$, the integration needs to be performed along lines having a different tilting $\arctan(-\alpha \left(z^{\prime }\right)/\beta \left(z^{\prime }\right))$ in the $\left(x_{a},x_{b}\right)$ plane. As we shall see, the point-to-point correspondence between sample and detector points is not altered by finite optical components, but other features of the reconstructed images are, such as the field of view and signal-to-noise ratio. In the following sections, expressions clarifying the role of apertures and detector size will be provided, as well as a strategy aimed at eliminating the artifacts they induce. The difference between the ideal case of a continuous correlation function and the experimental reality, in which only discrete values are available because of the pixel size, will also be explored and exemplified through simulations.

## 3. Results

### 3.1. Effect of the Finite Apertures

Due to the independence of α and β in Equation (Equation 5), on the transverse coordinates, the finite size of apertures within the experimental setup does not influence the point-to-point correspondence between the object and the detector planes. Still, the finite size of the optical components determines both the maximum resolution of the focused image, as in conventional imaging, and the spatial extension of the correlation function in the $\left(x_{a},x_{b}\right)$ plane. Hence, despite the fact that finite apertures do not influence the resolution curves of CPI far from the focused planes [18,21,23], large apertures are still convenient to maximize the amount of collected information, and have important consequences on the signal-to-noise ratio of the final image [27].

To better see these points, we shall assume only one limiting aperture along each of the two optical paths going from the source to the detectors; all other apertures, if any, can be disregarded. In this approximation, the function Φ can can be expressed as:(7)$$\Phi \left(x_{s},x_{s}^{\prime },x_{a},x_{b}\right)=\int_{-\infty }^{+\infty }\int_{-\infty }^{+\infty }\phi \left(x_{s},x_{s}^{\prime },x_{l},x_{l}^{\prime }\right)P_{A}\left(x_{l}\right)\varphi_{A}\left(x_{l},x_{a}\right)P_{B}\left(x_{l}^{\prime }\right)\varphi_{B}\left(x_{l}^{\prime },x_{b}\right)\;dx_{l}^{\prime }\;dx_{l},$$
with $\phi ,\;\varphi_{a}$, and $\varphi_{b}$ functions propagating the fields to and from the planes of the lenses, that vary with the specific CPI configuration. $P_{A,B}$ are the pupil functions of the limiting apertures in the optical paths leading towards the detectors $D_{A,B}$. By combining Equations (Equation 4) and (Equation 7), and solving all the integrals through a stationary phase approximation (geometrical optics limit), the correlation function can be expressed as:(8)$$\Gamma_{\mathrm{fin}}^{\left(2\right)}\left(x_{a},x_{b}\right)=\Gamma_{\mathrm{geom}}^{\left(2\right)}\left(x_{a},x_{b}\right){\left|P_{A}\left(a_{1}x_{a}+a_{2}x_{b}\right)\right|}^{2}{\left|P_{B}\left(b_{1}x_{a}+b_{2}x_{b}\right)\right|}^{2}.$$

Notably, in the geometrical approximation, the intensity transmission functions of the lenses (or any other limiting aperture) appear as multiplying factors in the correlation function. The arguments of the pupils are linear combinations of the detector coordinates, with coefficient $a_{1,2}$, in $P_{A}$, and $b_{1,2}$, in $P_{B}$, as for the object transmissivity. Unlike the coefficients α and β that appear in the object transmissivity, however, the new coefficients $a_{1,2}$ and $b_{1,2}$ are independent of the axial position of the object *z*; hence, the location of the object does not modify the areas enclosed in the transmissive region of the pupils.

In summary, Equation (Equation 8) indicates that the correlation function is still populated by a set of lines, each one corresponding to an object point, and whose tilting is determined by the coefficients α(z) and β(z), as in the case of infinite apertures (see Equation (Equation 5)); the novelty here is that these lines are limited in extension by an object-independent area that is defined by the finite size of the apertures.

### 3.2. Effect of the Finite Size of the Detectors

The limited extension of the photosensitive surface of the detectors is taken into account by imposing that the light intensity distribution at the detector planes is zero outside the photosensitive region. This means that substituting:(9)$$I_{I}\left(x_{i}\right)⟶\chi_{d_{i}}\left(x_{i}\right)I_{I}\left(x_{i}\right)$$
into Equation (Equation 1), with I=A,B; i=a,b; $d_{i}$ the linear size of the photosensitive surface of the two detectors, and $\chi_{d_{i}}$, the characteristic function of the photosensitive region, that equals 1 when its argument falls within a $d_{i}$-long segment centered on 0, and zero elsewhere. The effect of the characteristic functions of the detectors is to further reduce the area of the non-vanishing part of the correlation function, which now reads:(10)$$\Gamma_{\mathrm{fin}}^{\left(2\right)}\left(x_{a},x_{b}\right)=\Gamma_{\mathrm{geom}}^{\left(2\right)}\left(x_{a},x_{b}\right)\prod_{J,j}\chi_{d_{j}}\left(x_{j}\right){\left|P_{J}\left(j_{1}x_{a}+j_{2}x_{b}\right)\right|}^{2}.$$

Equation (Equation 10) indicates that the correlation function is now a tilted and re-scaled version of the correlation function obtained in the geometric approximation for infinite apertures and sensors ($\Gamma_{\mathrm{geom}}^{\left(2\right)}$ defined in Equation (Equation 5)), provided the coordinates $\left(x_{a},x_{b}\right)$ are both within the photosensitive areas of the detectors, and the ray corresponding to this pair of coordinates crosses the planes of the finite apertures through their transmissive region. For all other values of the coordinates $\left(x_{a},x_{b}\right)$, the correlation function is zero.

To offer a graphical representation of the correlation function, we shall now consider a particular CPI scheme, named *CPI between arbitrary planes* (CPI-AP) [23]. The peculiarity of this configuration is that two planes in the proximity of the object are imaged by the two sensors at a diffraction-limited resolution. In Figure 2a, we report the specific single-lens realization of CPI-AP that will be employed in all simulations. Here, light emitted by a two-dimensional object, such as a fluorescent sample, propagates towards a lens whose focal length is f=100mm; after the lens, the light beam is split towards two detectors $D_{A}$ and $D_{B}$, endowed with spatial resolution. The two lens-detector distances are such that the conjugate plane of detector $D_{A}$ is at a distance $z_{a}=210\;\mathrm{mm}$ from the lens, with magnification $M_{A}=0.91$, and the conjugate plane of detector $D_{B}$ is at a distance $z_{B}=190\;\mathrm{mm}$ on $D_{B}$, with magnification $M_{B}=1.1$. The object, placed at a distance *z* from the lens, is modeled as an infinitely extended distribution of equally-spaced Gaussian apertures, having center-to-center distance equal to 500μm and width equal to 100μm, as reported in Figure 2b. The size of the detectors is $d_{a}=d_{b}=10\;\mathrm{mm}$.

In the geometrical approximation, the correlation function of the apparatus described above reads: (11)$$\begin{array}{cc} \Gamma_{\mathrm{AP}}^{\left(2\right)}\left(x_{a},x_{b}\right) & ={\left|A\left(\alpha x_{a}+\beta x_{b}\right)\right|}^{4}{\left|P_{l}\left(ax_{a}+bx_{b}\right)\right|}^{4}\chi_{d_{a}}\left(x_{a}\right)\chi_{d_{b}}\left(x_{b}\right), \end{array}$$(12)$$\begin{array}{cc} & \begin{array}{cc} \alpha =\frac{1}{M_{a}}\frac{z-z_{b}}{z_{a}-z_{b}} & \beta =-\frac{1}{M_{b}}\frac{z-z_{a}}{z_{a}-z_{b}}\\ a=-\frac{1}{M_{a}}\frac{z_{b}}{z_{a}-z_{b}} & b=\frac{1}{M_{b}}\frac{z_{a}}{z_{a}-z_{b}} \end{array}. \end{array}$$
were A(x) is the aperture function of the object and ${\left|P_{l}\left(x\right)\right|}^{2}=\exp\left(-x^{2}/2l^{2}\right)$ is the pupil function of the lens. The coefficients α and β, on which the image of the object depends, determine the tilting of the integration path required for refocusing. In this particular CPI implementation, the two limiting apertures in the optical paths (see Equation (Equation 10)) depend on the same linear combination of the detector coordinates; the coefficients *a* and *b* determine the non-zero area of the correlation function, defined by the lens and detector size. In Figure 3a, we report the plot of the simulated correlation function, showing the effect of the finite size of the lens aperture. The simulation has been carried out in the paraxial approximation by considering a Gaussian lens of aperture *l*, whose numerical aperture (NA) shall be estimated as $\sin\left(l/f\right)$, and a perfect point-like correlation (i.e., infinitesimal coherence area) on the source. The geometrical approximation has not been applied, and we will show that, even though all the data processing considerations that have been and will be discussed and demonstrated are based solely on geometrical arguments, they work equally well outside of such approximation. The plots in Figure 3a show that, as the numerical aperture has increased from 0.051 to 0.25, the extension of the non-zero portion of the correlation function, within the $\left(x_{a},x_{b}\right)$ space defined by the detectors, greatly increases. The region of the $\left(x_{a},x_{b}\right)$ space enclosed within the two red dashed lines represents the lens extension, which we have chosen as the portion of space for which ${\left|P_{l}\left(x\right)\right|}^{2}>0.001$. Regardless of the particular value of the threshold selected to identify the lens area, something very interesting can be inferred from the plots. Even without the geometrical optics approximation, the finite lens aperture sets a boundary to the areas where correlations can be measured: correlations are non-zero as long as $\left|ax_{a}+bx_{b}\right|\leq c\cdot l$, where *c* is a constant chosen to define the size of the Gaussian lens. The finite lens aperture also influences the field of view (FOV), which can be thought as the number of slits that contribute to the measurable correlations. Since the area $\left|ax_{a}+bx_{b}\right|\leq c\cdot l$ is infinitely extended, each point $x_{s}$ of the object gives rise to a non-zero contribution of the correlation function along the segment defined on the line:(13)$$\alpha x_{a}+\beta x_{b}=x_{s}$$
and enclosed by the two lines of equations:(14)$$ax_{a}+bx_{b}=\pm c\cdot l.$$

Interestingly, since the matrix:(15)$$\left(\begin{array}{cc} \alpha & \beta \\ a & b \end{array}\right)$$
obtained from the coefficients in Equations (Equation 13) and (Equation 14) is never singular, such a segment is always well defined. However, the only object details that contribute to the measured correlation function are those associated to segments $d_{a}$ and $d_{b}$, in the $\left(x_{a},x_{b}\right)$ plane, that are (at least partly) contained within the photosensitive portion of the detectors, namely, segments containing points such that $\left|x_{a}\right|\leq d_{a}/2$ and $\left|x_{b}\right|<d_{b}/2$. The length of the segment under consideration defines the number of “replicas” that a given object point produces in the correlation function. This gives rise to an important difference of CPI with respect to conventional imaging, where the FOV (in the absence of aberrations) is only defined by the detector size. In fact, as we shall see in the next section, the length of the segment in the space defined by the detector coordinates plays an important role in determining the successful reconstruction of the final images, particularly in terms of their signal-to-noise ratio [18]. In fact, the combined effect of the lens and the detector size is to shorten the length of the segment; at some point, the segment will be too short for the associated object detail to be effectively reconstructed. Interestingly, given the chosen optical parameters and detector size, the extension of the lines associated with the object coordinates is almost exclusively limited by the detector size already for NA = 0.25; hence, a larger numerical aperture would yield no advantage in that regard.

Figure 3b shows how the position of the object along the optical axis influences the inclination of the segments mentioned above, as given by Equation (Equation 13). When the object is placed on either one of the two planes that are conjugate to the detectors ($z=z_{a}$ or $z=z_{b}$), the object details line up along the horizontal ($z=z_{b}$) or along the vertical ($z=z_{a}$) and the values of the coefficients α and β are (0,1) and (1,0), respectively. This result is rather intuitive: when $z=z_{a}$ ($z_{b}$), the slits are on focus on detector $D_{A}$ ($D_{B}$) (i.e., along the $x_{a}$- ($x_{b}$-)axis) independent of the specific coordinate $x_{b}$ ($x_{a}$) on detector $D_{B}$ ($D_{A}$). The segments associated with object points are tilted when the object is placed in any other plane different from $z_{a}$ and $z_{b}$. In particular, the replicas of the object points are arranged along a line that forms an angle θ with the $x_{a}$-axis such that $\tan\theta =\frac{M_{b}}{M_{a}}\frac{z-z_{b}}{z-z_{a}}$. This can be better understood by analyzing Figure 4a, where we report the correlation function associated with an object placed at z=195mm (Figure 3a, center).

The measurable correlation function is contained in the region given by the intersection between the area of the lens (enclosed by the blue lines) and the area defined by the two sensors (orange square). All segments corresponding to a given object point $x_{s}$ identify a preferred direction (red dashed line) along which the correlation function can be integrated, so as to reconstruct the object without losing resolution (i.e., to refocus it). This preferred direction is at the expected $\theta =-18.4{}^{\circ }$. Depending on the value of the coordinate $x_{s}$, the corresponding segment can:1.Fall outside of the area defined by the detectors for slits that are outside the FOV;2.Fall only partially outside of the detector area so that, upon reconstruction, the corresponding slits will receive a decreasing amount of contributions than that available on the lens area (red area);3.Be fully included in the detector area, thus enabling reconstruction of the corresponding object point with all possible contributions from the lens (green area).

As we shall see below, if these different conditions are not correctly accounted for, upon refocusing, artifacts will appear in the reconstructed image. However, once again, the geometrical optics approximation (Equation (Equation 10)) is an adequate tool for quantifying these effects and finding a strategy to eliminate the artifacts. In fact, by applying the refocusing process described by Equation (Equation 6), the correlation function of Equation (Equation 8) gives the product of the object aperture transmissivity with the function:(16)$$\Sigma_{\mathrm{corr}}\left(x_{s}\right)=\int_{\gamma_{z}\left(x_{s}\right)}\prod_{J,j}\chi_{d_{j}}\left(x_{j}\right){\left|P_{J}\left(j_{1}x_{a}+j_{2}x_{b}\right)\right|}^{2}d\ell ,$$
where $\gamma_{z}\left(x_{s}\right)$, defined by the equation $\alpha \left(z\right)x_{a}+\beta \left(z\right)x_{b}=x_{s}$, is the integration line associated with the object detail identified by the coordinate $x_{s}$. This gives rise to the artifacts mentioned above. However, the aperture function can still be properly reconstructed by either dividing the refocused image of Equation (Equation 6) by the function in Equation (Equation 16), or by dividing the non-zero part of the correlation function by the integrand function of Equation (Equation 16), prior to applying the refocusing algorithm. Either way, the effect of the different length of the segments contributing to different parts of the sample can be factored out, enabling the proper reconstruction of the object, as shown in Figure 4b. Here, the blue curve represents the infinitely extended set of Gaussian apertures composing the sample. The effect of the finiteness of the optical components is evident in the uncorrected refocused image (black curve): near the edges of the FOV, defined by the finite sensor sizes, the slits receive less contributions due to the shorter extension of the integration segments. This effect, which depends on the angle of inclination of the integration line (determined by the axial position *z*), can be compensated by applying the proposed correction. The effectiveness of this correction strictly relies on how well the theoretical model, bringing to the expression in Equation (Equation 16), adheres to the experimental situation. In order to experimentally evaluate the correction function and calibrate the device, one would gain access to the integrand function of Equation (Equation 16). This could be conducted by measuring correlations after either removing the sample, if working in transmission, or replacing it with a mirror, if working in reflection.

### 3.3. Effect of the Finite Size of the Pixels

From a formal point of view, the effect of the pixel size ($\delta x_{a,b}$) on the planes of the sensors $D_{A}$ and $D_{B}$ implies that the measured 2D correlation function is described by the quantity:(17)$$I_{I}\left(x_{i},t_{j}\right)=\int_{x_{i}-\delta x_{i}/2}^{x_{i}+\delta x_{i}/2}\int_{t_{j}}^{t_{j}+\Delta t}I_{I}\left(x_{i}^{\prime },t\right)dt\;dx_{i}^{\prime }$$
rather than by Equation (Equation 2). The spatial coordinates vary within the discrete set associated with the pixel centers, on the two detectors. The first obvious limitation that such discretization imposes to CPI is resolution, in a similar fashion as it does to conventional imaging. From the insight built-up so far, we know that two neighboring point-like details on the object plane, placed at coordinates $x_{s}$ and $x_{s}+\delta x_{s}$, will originate two lines of equations $\alpha x_{a}+\beta x_{b}=x_{s}$ and $\alpha x_{a}+\beta x_{b}=x_{s}+\delta x_{s}$ in the $\left(x_{a},x_{b}\right)$ space where the correlation function is defined. This plane is available as a discrete 2D space with points placed in a regular lattice with spacing $\delta x_{a}$ along the horizontal and $\delta x_{b}$ along the vertical directions. The spacing between the two lines associated with the two point objects is $\frac{\left|\delta x_{s}\right|}{\sqrt{\alpha^{2}+\beta^{2}}}$; when this value is comparable to the pixel size, discriminating between the two lines becomes impossible. A rough yet intuitive estimate of the minimum separation $\left|\delta x_{s}\right|$ that can be resolved is obtained by imposing that the two lines never cross the same $\delta x_{a}\cdot \delta x_{b}$ cell in the $\left(x_{a},x_{b}\right)$ discrete space. This is expected to depend both on the angle $\theta =\arctan\left(-\alpha /\beta \right)$ of inclination of the lines and on the size of the pixels of the two detectors. Geometrical considerations (reported in Appendix A) lead to the following condition for the minimum separation $\delta x_{s}\left(z\right)$:(18)$$\delta x_{s}\geq \delta x_{a}\left|\alpha \right|+\delta x_{b}\left|\beta \right|.$$

This expression reduces to the one known for conventional imaging when the object is on focus ($z=z_{a,b}$), that is, when $\left(\alpha ,\beta \right)=\left(1/M_{A},0\right)$ or $\left(\alpha ,\beta \right)=\left(0,1/M_{B}\right)$, where $M_{A}$ and $M_{B}$ are the magnifications on the conjugate planes of detectors $D_{A}$ and $D_{B}$. Unlike conventional imaging, however, Equation (Equation 18) implies a dependence of the resolution on the axial position *z* of the unfocused object. In particular, the coefficients α and **β** can diverge at large distances from the focused planes, as in the example chosen for the simulations. Similar to conventional imaging, both the pixel size and the finiteness of the point-spread function imposes limits on the maximum achievable resolution. A plot of the pixel-limited resolution, normalized over the pixel size (supposed to be equal on the two detectors) is shown in Figure 5a.

Figure 5b displays the effect of pixelization by considering the same object and setup as in the previous section, and choosing a pixel size of $\delta x_{a}=\delta x_{a}=200\;\mu m$. In this condition, based on Equation (Equation 18), an object with a spacing of 500μm is expected to produce lines that are separated by roughly two pixels in the $\left(x_{a},x_{b}\right)$ plane, in good agreement with the result shown in the figure.

The importance of retrieving multiple shifted “replicas” of the same object portion within the measured correlation function becomes particularly relevant when considering noise. In fact, a typical drawback of correlation-based imaging, is the low SNR of the measured correlation function. CPI makes no exception: in a typical scenario, the signal-to-noise ratio is lower than unity, as shown in Figure 5c, where details of the object are not immediately recognizable due to noise burying the space modulation on the correlation function induced by the object features. The image shown here was obtained by superimposing a white-noise mask to the pixelated correlation function of Figure 5b, in such a way to obtain an SNR of 0.5 at best. Due to the addition of noise, Pearson’s correlation coefficient between Figure 5b,c drops to 42.7%; nevertheless, upon refocusing, the correlation with the reference object was 72.1%, showing that the sample features were reconstructed even without the need for noise suppression algorithms. These conditions are ideal for showing the importance of retrieving multiple “replicas” of each object detail: despite the noise and pixelization limit the number of contributions to the same object coordinate, the process of refocusing, by making use of the redundancy of the correlation function, can (at least partially) isolate the object features from the noise background. Larger apertures and detectors, combined with smaller pixels, are thus always beneficial to CPI, as, besides improving the FOV, also increase the quality of the final images, thanks to statistical effects.

## 4. Discussion

The effect of the finite size of the optical components is difficult to treat mathematically due to the fact that, in most cases of interest, Equation (Equation 7) cannot be solved analytically. However, a sufficiently deep insight can be gained in the geometrical optics context. The predictions based on the geometrical approach have proven to be quite reliable, and show that finite apertures do not alter the geometrical point-to-point correspondence between object and detector points. Their main effect is to limit the extension of the measurable correlation function in the 4D space defined by the coordinates of the two detectors. Interestingly, this limitation is independent of the position of the object along the optical axis. A similar result was found when accounting for the finite photosensitive area of the detectors: non-zero contributions to the correlation function were experimentally available only as long as the reconstructed optical rays crossed the finite apertures in their transmissive regions and were detected within the sensor surface. The combined effect of finite detectors and apertures is to limit the FOV of the reconstructed image, as well as to create artifacts; the latter can be eliminated either through accurate theoretical modeling or by experimental calibration of the device.

We have also found that the finite size of the pixels on the detectors imposes limitations on the achievable resolution of the refocused image, and this occurs in a way that strongly depends on the position of the object along the optical axis. The pixel size also determines how many times the same detail on the object plane is encoded in the correlation function. Being based on the superposition of redundant details, the refocusing procedure is much more efficient, in terms of noise suppression, when the object details are available with a higher redundancy.

Future developments on the topic shall include optimization of the data processing algorithm to the case of the experimental discrete correlation function. Applying the refocusing procedure and other 3D reconstruction techniques can be demanding in terms of time and computational resources; hence, in conjunction with the use of next generation ultra-fast sensors [28], an approach requiring the minimum amount of resources is needed in view of applications such as real-time imaging.

## Figures and Tables

**Figure 1 sensors-22-02778-f001:**
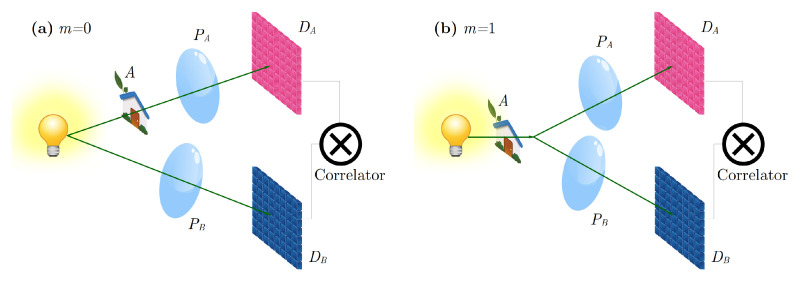
Comparison between different CPI protocols. (**a**) The source can emit either entangled photons or chaotic light. Light from the source is split into two paths: in one arm, light illuminates the object, passes through a lens and is detected by detector $D_{a}$; in the second arm, light propagates freely toward a lens and is collected by sensor $D_{b}$. in Equation (Equation 4). (**b**) The source emits chaotic light, which illuminates the object, and is then split into two paths; each beam then passes through a lens and is collected by a detector. In both panels, *A* indicates the aperture function of the object, $P_{A,B}$ the pupil functions of the lenses, $D_{A}$ and $D_{B}$ are high-resolution detectors, ⊗ indicates that correlations are measured between the two detectors $D_{A}$ and $D_{B}$.

**Figure 2 sensors-22-02778-f002:**
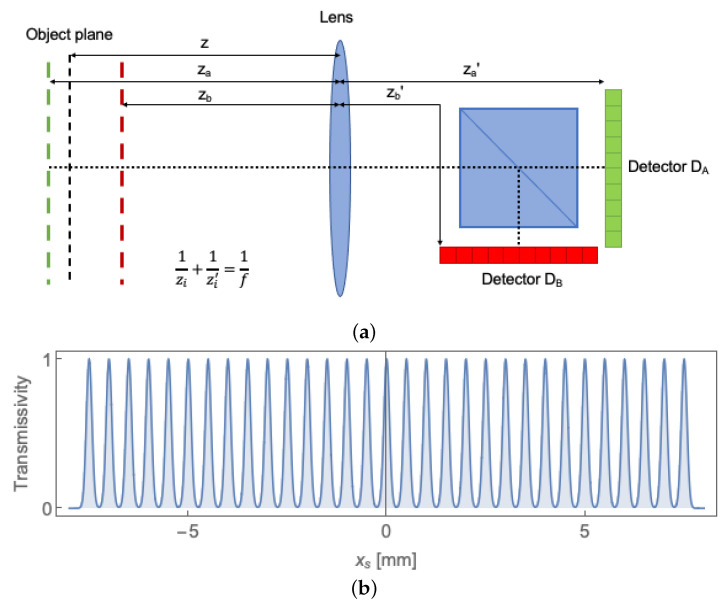
(**a**) Example of a CPI setup, named *correlation plenoptic imaging between arbitrary planes* [23]. A source of thermal light is placed on the object plane, and plays the role of both the illuminating source and the object (e.g., a fluorescent object). Photons from the object propagate through a single lens with focal length *f* and, after being separated by a beam-splitter, are collected by two spatially-resolving detectors $D_{A}$ and $D_{B}$. The planes optically conjugated to the detectors do not coincide ($z_{a}\neq z_{b}$) and, in the most general case, the object is placed out of focus with respect to both detectors ($z\neq z_{a,b}$). Details about the experimental parameters can be found in the text. This setup is used to generate all the simulations reported in this work. (**b**) The intensity transmissivity of the object used for the simulations (${\left|A(x_{s})\right|}^{2}$). The object is an indefinitely extended set of equally-spaced (500μm) Gaussian apertures of width 100μm.

**Figure 3 sensors-22-02778-f003:**
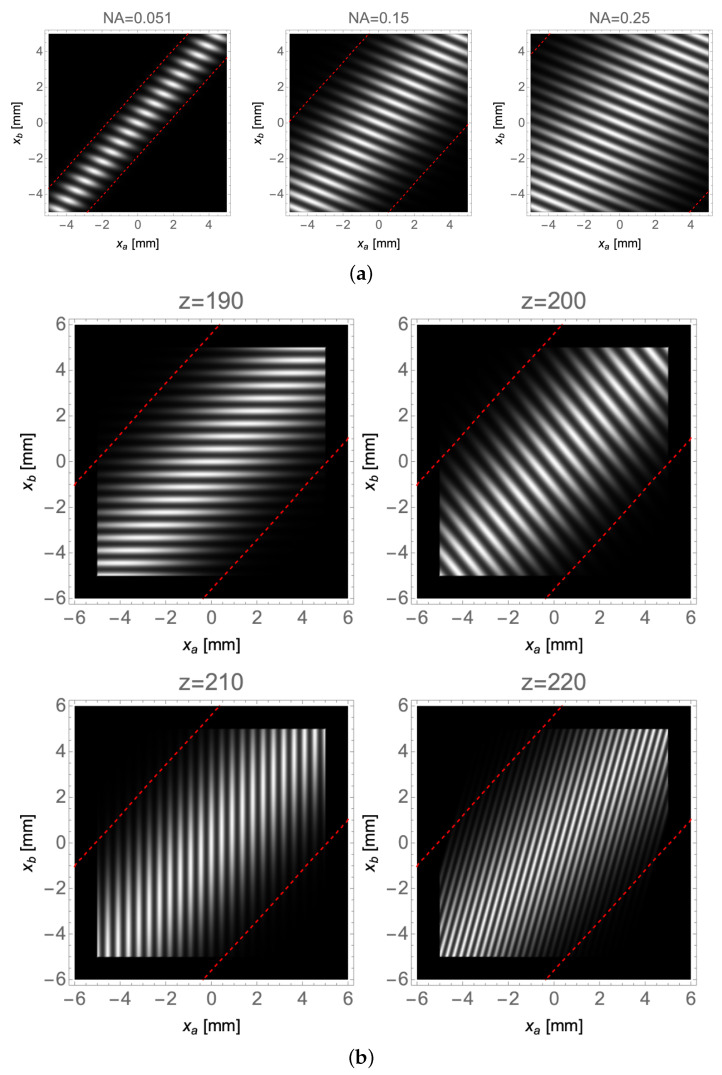
(**a**) Study of the effect of the numerical aperture (NA) of the lens on the extension of the correlation function in the $\left(x_{a},x_{b}\right)$ plane. The numerical aperture is changed by varying the lens aperture *l*. With reference to Figure 2a, the object was placed at a distance $z=195\;\mathrm{mm}\neq z_{a,b}$ from the lens. The area enclosed between the red dashed lines represents the lens extension (in this region, the lens aperture function, whose values vary in (0,1), is larger than 0.001). (**b**) The effect of the displacement of the object along the optical axis, at a fixed lens numerical aperture of 0.15. Four axial positions of the object were explored. When the sample was placed in either one of the two planes of focus ($z=z_{a}=190\;\mathrm{mm}$ and $z=z_{b}=210\;\mathrm{mm}$), the replicas of the details appearing in the correlation function line up either horizontally or vertically. When z=200mm the object was placed between the two planes focused on $D_{A}$ and $D_{B}$; while z=220mm corresponds to a sample placed farther away from the lens than the farthest plane focused on $D_{A}$ and $D_{B}$.

**Figure 4 sensors-22-02778-f004:**
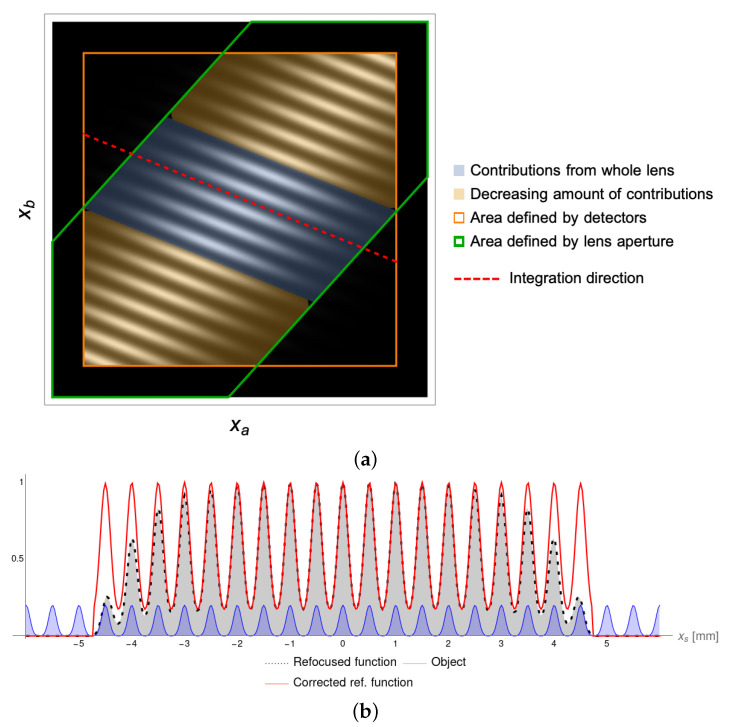
(**a**) Subdivision of the correlation function into parts characterized by different properties. The strip enclosed between the two green lines is the area defined by the lens aperture, and the squared area defined by the orange line identifies the space defined by the size of two detectors. The correlation function can only be measured in the region defined by the overlap of the two. The red dashed line identifies the integration direction, namely, the direction along which the correlation function contains information about the detail located at coordinate $x_{s}$ on the object plane. As the integration line sweeps the entire object plane (i.e., all possible values of $x_{s}$), two cases can occur: either the same detail receives contributions from the whole lens or, due to the finite size of the detectors, contributions decrease towards the edges. (**b**) Effect of the finite size of the lens and the detectors on the reconstruction of the object. The solid blue line represents the reference object, in arbitrary units. Points of the correlation function located close to the edge of the detectors area (red area in (**a**)) give rise to artifacts in the reconstructed object (black dashed line), due to the varying amount of contributions received by the corresponding object details. Artifacts can be removed by applying the correction discussed in the text Equation (Equation 16), so that all the slits within the field of view can be reconstructed (solid red line) with their correct relative height.

**Figure 5 sensors-22-02778-f005:**
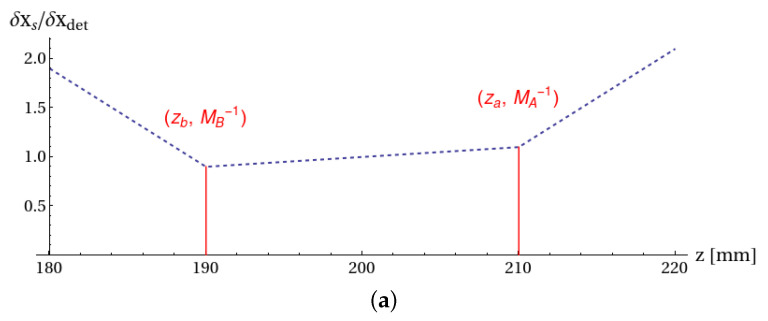
Analysis of the effect of pixelization on the correlation function. Pixel size was taken to be 200μm. (**a**) Resolution curve due to pixelization. When the object was placed between the two planes at focus, the resolution defined by the pixel size was given by the linear interpolation between the points $\left(z_{a},M_{A}^{-1}\right)$ and $\left(z_{b},M_{B}^{-1}\right)$. If the object was placed outside of the two planes, the required pixel size needed to discriminate that close details rapidly decrease. (**b**) Effect of pixelization on the correlation function when the object was placed out of focus (at z=195mm from the lens); the pixel size (200μm) corresponds to 50 pixels per detector. (**c**) Typical aspect of a measured correlation function: zero-average white noise has been superimposed on the normalized theoretical correlation function, so as to simulate a correlation function with a signal-to-noise ratio, which was, at best, 0.5 (standard deviation of white noise amounts to 2 arbitrary units). (**d**) Refocused image obtained from the correlation function reported in panel (**c**) (blue line), as compared with the reference object (orange line). The refocused image was obtained by applying the refocusing algorithm in Equation (Equation 6) and dividing by the correction coefficient in Equation (Equation 16).

## Data Availability

Not applicable.

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
