# Peer review of "Effect of Finite-Sized Optical Components and Pixels on Light-Field Imaging through Correlated Light"

_sensors, 2022, doi:10.3390/s22072778_

Round 1

Reviewer 1 Report

Please find my comments in the report attached.

Author Response

We thank the Reviewer for appreciating our work, recommending it for publications after minor changes, and giving us detailed suggestions for improving its clarity and completeness. Please find below our reply (black) to all Reviewer comments (reported in blue):

The Authors report on the effect that of both the finite extension of optical components and detectors, and the discrete nature of the measuring process (pixel size), have in a correlation plenoptic imaging (CPI) setup. They focus on how these parameters affect the measured intensity correlation function with special attention to the refocusing capability of the CPI system. The article is simple, but I find it valuable because it accounts for practical aspects that have not been addressed so far that lead to useful conclusions. I recommend the publication of the paper after a minor revision of the manuscript.

See my comments below.

The description of the state of the art is too succinct. Specifically, besides CPI the Authors only mention the traditional PI arrangement where a microlens array is introduced between the sensor and the imaging device. There exist several other approaches, techniques, and arrangements to perform light-field imaging. The Authors may find useful the references below as well as the references therein and more recent work citing them. Also, Ref. [16] should be better used and serve this purpose.

Birklbauer & Bimber (2014, May). Panorama lig Computer Graphics Forum (Vol. 33, No. 2, pp. 43-52).
Xue, et al., (2014, March). Light field panorama by a plenoptic camera. In Computational Imaging XII (Vol. 9020, 186-196). SPIE. Ihrke, et al., (2016). Principles of light field imaging: Briefly revisiting 25 years of research. IEEE Signal Proc. Mag., 33(5), 59-69. Zhu, et al., (2017). Light field imaging: models, calibrations, reconstructions, and applications. Front. Inform. Tech. El., 18(9), 1236. Lien, et al., (2020). Ranging and light field imaging with transparent photodetectors. Nature Photonics, 14(3), 143-148.
We thank the referee for suggesting the above references, which have all been included in the bibliography and contribute to improve the characterization of the state of the art concerning light-field imaging.

I strongly suggest making a figure for Section 2, to graphically illustrate the formalism and make it easier for the reader to follow the meaning of the different terms that appear in the equations.
We thank the reviewer for suggesting the addition of a figure. The revised version of the manuscript contains the newly added Figure 1, in which the meaning of terms appearing in the equations is graphically clarified.

Page 3, line 82: I believe the sentence should read An analogous approximation holds for the case k = 1 in Eq. (1) . Of not, please clarify then meaning of such sentence.
The sentence has been revised according to the reviewer’s suggestion.

Around Eq. (5), please clarify if the aperture function is the same as the two-dimensional, polarization- independent amplitude transmissivity of the object.
The aperture function A is the same as in Equation (4). We have clarified this point in the text.

What is the meaning of the integer coefficient in Eq. (4-5)? When does it take a value of 0 or 1?
The integer coefficient m depends on the realization of Correlation Plenoptic Imaging: if the object is placed only in one of the two arms, then m = 0; if light illuminates the object before it is split towards the detectors, then m = 1. We have clarified this point in Section 2, corroborated by the newly added Figure 1.

How are the coefficients and appearing in Eq. (5) calculated? Why are these coefficients functions only of the axial position and have no transverse dependence?
The coefficients in Equation (5) are calculated by applying a stationary phase method for evaluating the integrals in Equation (4). A clarification has been added in the text just above Equation (5).
The coefficients are a function of the experimental parameters that appear in the analytical expression of $\Phi$, such as the axial coordinate. Being coefficients of a linear combination of the transverse coordinates, they cannot depend on the transverse coordinates themselves. We have clarified the reason behind this linear combination also in the lines preceding Equation (5).

Why are the coefficients in Eq. (8) functions only of the transverse position?
The same considerations in the reply to the previous point hold in this case.

Makes sure the phrase correlation plenoptic imaging does not appear after the abbreviation CPI has been introduced. I suggest introducing the abbreviation ICF for intensity correlation function .
We have fixed the point concerning CPI. We have decided to avoid using the abbreviation ICF due to partial inaccuracy of the phrase “intensity” correlation function, since the discussed quantity is, in the case of chaotic light, the correlation between intensity fluctuations (covariance of the intensities, not mere correlation).

The English language must be thoroughly revised. There are many typos and grammar mistakes throughout the manuscript.
The text has been reviewed in this regard.

Reviewer 2 Report

In this manuscript, authors analyze the conditions of the pixel-limited resolution of the refocused images, as well as a strategy for eliminating artifacts in the finite-sized optical components. Mathematical analysis and approach looks fine. English grammar looks fine.However, it is a kind of being hard to understand what the proposed idea is useful. Therefore, authors need to emphasize the novelty of the proposed idea. In addition, reviewer does not understand why proposed idea could be published. Therefore, authors need to address these points as below.

1. Authors need to provide some more recent literature review in the introduction section.
2. In Equation (12), what are alpha, beta, a, and b?
3. Please change Eq. to Equation.
4. In Figure 3 (a), it is still hard to distinguish 4 areas.
5. Data availability section is missing ?
6. Please use abbreviated journal names in the reference sections.
7. Only 2 keywords might be small in Line 14.
8. In Figure 2, it is hard to see what x- and y-axes labels are because of unclear fonts.
9. There are no specific output data as you saw in Figure 4. Authors need to make a Table to show some performance such as intensity and aberration.
10. In Figure 2, authors did not show the image for the NA over 0.5.

Author Response

We thank the reviewer for the overall positive assessment of our manuscript and giving us detailed suggestions for improving its clarity and completeness. Please find below our reply (black) to all Reviewer comments (reported in blue):

In this manuscript, authors analyze the conditions of the pixel-limited resolution of the refocused images, as well as a strategy for eliminating artifacts in the finite-sized optical components. Mathematical analysis and approach looks fine. English grammar looks fine.However, it is a kind of being hard to understand what the proposed idea is useful.Therefore,

Authors need to emphasize the novelty of the proposed idea. In addition, reviewer does not understand why proposed idea could be published. Therefore, authors need to address these points as below.
We have improved the discussion in the introduction to motivate the novelty of the work and its fundamental and practical relevance in the context of Correlation Plenoptic Imaging.

  1. Authors need to provide some more recent literature review in the introduction section.
    We have added to the bibliography references containing recent developments.
  2. In Equation (12), what are alpha, beta, a, and b?
    The coefficients in Equation (12) are those introduced in Equation (11). Latex formatting had made so that the two were separated by two pages containing figures, and that was unfortunate. The problem has been fixed with an improved layout.
  3. Please change Eq. to Equation.
    We have followed the reviewer’s suggestion and made such a change.
  4. In Figure 3 (a), it is still hard to distinguish 4 areas.
    The figure has been improved with thicker lines and different colors.
  5. Data availability section is missing?
    There is no data associated with this manuscript, besides those directly shown in the plots. If the editor deems it necessary, we will add such a statement in the manuscript.
  6. Please use abbreviated journal names in the reference sections.
    We use abbreviated journal names in the revised version of the manuscript.
  7. Only 2 keywords might be small in Line 14.
    We have extended the list, which now contains four keywords.
  8. In Figure 2, it is hard to see what x- and y-axes labels are because of unclear fonts.
    Figure 3 (which was Figure 2 in the first version) has been redrawn and font size changed.
  9. There are no specific output data as you saw in Figure 4. Authors need to make a Table to show some performance such as intensity and aberration.
    We thank the reviewer for pointing out the need to quantify the improvement in image quality. In the case of added noise, we have quantified the effect of noise on the correlation function by evaluating Pearson’s correlation between the noiseless function and the one with added noise. The quality of the reconstruction after refocusing has been evaluated by correlating the refocused function and the reference object.
  10. In Figure 2, authors did not show the image for the NA over 0.5.
    Given the choice of parameters, particularly the detector size, any NA larger than those shown in the manuscript would have no effect on the correlation function. This point has been clarified in the revised text.

Reviewer 3 Report

This paper introduced effects of finite extension of optical components detectors, and pixel size on the measured correlation function and the refocusing capability of CPI, where an expression was provided to evaluate the pixel-limited resolution of the refocused images. My concerns are as follows:
1. Too much background introductions in the abstract. After all, the abstract should focus on the work of this paper and its innovativeness. 
2. Equation problems.
In Page 3, eq(4), please explain what the 'A*(xs'), xs, xs' means to make it clearer presentation.
In Page 3, eq(5), the leter 'm' should be explained in detail.
In page 3, Line 119, on the right side of equation “ A(αxa + βxb ) = A(αx xa + βx xb, αyya + βyyb)”, what do the 'xa , ya, xb, yb' mean?
In Page 3, Line 124, the equations should be explained in more details.
3. Section 2, the method of this paper and its innovativeness lack a detailed description. The paper describes the principle of correlation imaging, which consists of linear addition of two detectors. However, it does not introduce the use of algorithms to eliminate the influence of the size and pixel of optical elements on imaging results, which is not conducive to understanding the subsequent experimental results.
4. Section 3, the influences of limited aperture, limited detector size and pixel size are respectively discussed. In the first section, the comparison of imaging effects under different aperture sizes is lacking when the influence of aperture size limits on images is introduced. 
5. In Page 4, eq(8), How  eq(8) is obtained combing the eq(7) and eq(4), it should be explained clearly here. 
6. Some images in the paper are not clear, such as Figure 2.
7. English spell check is required.

Author Response

We thank the Reviewer for appreciating our work and giving us detailed suggestions for improving its clarity and completeness. Please find below our reply (black) to all Reviewer comments (reported in blue):

This paper introduced effects of finite extension of optical components detectors, and pixel size on the measured correlation function and the refocusing capability of CPI, where an expression was provided to evaluate the pixel-limited resolution of the refocused images. My concerns are as follows:

  1. Too much background introductions in the abstract. After all, the abstract should focus on the work of this paper and its innovativeness.
    We thank the reviewer for this suggestion. We have shortened and modified the abstract to better focus on the presented work and its innovations
  2. Equation problems.

In Page 3, eq(4), please explain what the 'A*(xs'), xs, xs' means to make it clearer presentation.
We have clarified this point in the revised text.

In Page 3, eq(5), the leter 'm' should be explained in detail.
The meaning of the letter ‘m’ is now addressed in the text, after Eq. (4), and also explained with the newly added Figure 1.

In page 3, Line 119, on the right side of equation “ A(αxa + βxb ) = A(αx xa + βx xb, αyya + βyyb)”, what do the 'xa , ya, xb, yb' mean?
The pairs of variables (xa, ya) and (xb, yb) are the coordinates of the 2D planes corresponding to the detectors Da and Db, respectively. These definitions have been introduced at the beginning of Section 2. We have also changed notation to a more intuitive one, in which x and y are the components of a boldface variable $\rho$.
For clarity, we have removed the unclear expression mentioned by the referee and specified that, by construction, $\alpha $ and $\beta $ are the same for the two space directions x and y, and thus dropped the subscripts from these quantities.

In Page 3, Line 124, the equations should be explained in more details.
We have improved discussion of the equations by explaining in more detail the consequences of symmetry and factorization with respect to the coordinates. Such discussion can be found in lines 181-184.

  1. Section 2, the method of this paper and its innovativeness lack a detailed description. The paper describes the principle of correlation imaging, which consists of linear addition of two detectors. However, it does not introduce the use of algorithms to eliminate the influence of the size and pixel of optical elements on imaging results, which is not conducive to understanding the subsequent experimental results.
    We have improved the discussion in Sections 1 and 2 to account for the points raised by the reviewer. Our work is focused on properly quantifying and taking the effects of pixels and finite apertures and detectors into account; beside its physical fundamental relevance, it serves the purpose of offering the experimenter the criteria for optimizing the experimental setup. Introducing algorithms to eliminate or mitigate these effects is outside the scope of this work. The results shown in Figure 5d (Figure 4d in the first version of the manuscript) are obtained by only applying the typical refocusing algorithm defined in Equation (6), and accounting for the finite lens size. This aspect has now been clarified in the image caption.
  2. Section 3, the influences of limited aperture, limited detector size and pixel size are respectively discussed. In the first section, the comparison of imaging effects under different aperture sizes is lacking when the influence of aperture size limits on images is introduced.
    One of the distinguishing features of CPI is the fact that the refocused images are characterized by a transverse resolution that is essentially independent of the numerical aperture of the optical components, but only depends on the wavelength and involved longitudinal distances. This justifies why, in the manuscript, apertures and detector size are only analyzed in terms on their effect on the extension of the correlation function. We have added a discussion in the Introduction to explicitly clarify this point.
  3. In Page 4, eq(8), How  eq(8) is obtained combing the eq(7) and eq(4), it should be explained clearly here.
    Equations (7) and (4) are combined and involve a certain number of integrals that, typically, cannot be solved analytically and are approximated through a stationary phase method, as explained in section 1 (Reference also given in Section 1); the exact form of these integrals depends on the specific CPI setup, and is reported in all references discussing the corresponding CPI devices. This point is now clarified in the text.
  4. Some images in the paper are not clear, such as Figure 2.
    We have redrawn Figure 2 and 3a, which were unclear. They are now numbered 3 and 4a, due to the addition of a new Figure 1.
  5. English spell check is required.
    We have performed spell check and fixed typos.

Reviewer 4 Report

This paper presents a method to evaluate the pixel-limited resolution of refocused images.

The paper is interesting and sounds reasonable.

I recommend the authors give more focus on the need for this problem to be solved. Based on what I read, I didn't see the relevance of the topic.

Please, include more references to support the presented approach.

Moreover, review the references, some seem to be incomplete.

Author Response

We thank the reviewer for the overall positive assessment of our manuscript and giving us detailed suggestions for improving its clarity and completeness. Please find below our reply (black) to all Reviewer comments (reported in blue):

This paper presents a method to evaluate the pixel-limited resolution of refocused images.

The paper is interesting and sounds reasonable.

I recommend the authors give more focus on the need for this problem to be solved. Based on what I read, I didn't see the relevance of the topic.
We thank the reviewer for their comment. We have improved the discussion on the motivation of our work both in the Introduction and in the Materials and Methods section.

Please, include more references to support the presented approach.
We have added some references on general aspects of light-field imaging.

Moreover, review the references, some seem to be incomplete.
We have fixed the incomplete references.

Round 2

Reviewer 2 Report

Authors answered the questions so the paper can be recommended to be accepted.

Reviewer 3 Report

This manuscript has been revised based on review's comment. I think that this paper could be accepted, if the author improves quality of the paper as following.

  1. English spell check is required.
  2. Please recheck the format of references.  I recommend updating and using relevant high-level academic papers from the past ten years.